# Burden of Wasting and Stunting in Colombia and Its Economic Impact: A Society’s Perspective Analysis, 2021

**DOI:** 10.3390/nu16244302

**Published:** 2024-12-13

**Authors:** Darío Londoño Trujillo, Paula Andrea Taborda Restrepo, María Camila De la Hoz, Juan Carlos Burgos Castro, Joan Sebastian Arbelaez Vargas, Diana María Pineda Ruiz

**Affiliations:** 1Population Health Division, Fundación Santa Fe de Bogotá, Bogotá 110111, Colombia; dario.londono@fsfb.org.co (D.L.T.); maria.delahoz@fsffb.org.co (M.C.D.l.H.); 2School of Medicine, Unviersity of Andes, Bogotá 110111, Colombia; 3Faculty of Public Health, University of Antioquia, Medellín 050044, Colombia; 4Fundación Éxito, Medellín 050044, Colombia; t-jcburgos@grupo-exito.com (J.C.B.C.); t-jarbelaez@grupo-exito.com (J.S.A.V.); dpinedar@grupo-exito.com (D.M.P.R.)

**Keywords:** protein-caloric malnutrition, growth failure, burden of disease, cost of disease

## Abstract

Background/Objectives: Child malnutrition is a critical public health concern that significantly hampers children’s physical and mental development and imposes serious economic burdens. The World Health Organization (WHO) estimates that malnutrition is responsible for half of all deaths among children under five, leading to long-term consequences such as lower educational achievement, decreased productivity, and deepened poverty. This study aims to estimate the burden of child malnutrition in Colombia for children up to four years old, assessing both direct and indirect costs from a societal perspective. Methods: A Markov chain model was utilized to simulate six health states related to malnutrition, integrating direct and indirect costs. Epidemiological data and international literature informed the transition probabilities between states, while caregiver surveys were conducted to capture the indirect costs. Results: The study found that malnutrition accounted for 419.84 disability-adjusted life years (DALYs) per 1000 inhabitants. The total cost of malnutrition over a four-year period was approximately USD 243.58 million, with an annual average of USD 60.89 million, of which 65% of the burden fell on households. Conclusions: Child malnutrition in Colombia presents a considerable burden on health systems, households, and the national economy, demonstrating the need for robust interventions to mitigate its long-term socioeconomic impact.

## 1. Introduction

In 2015, the United Nations adopted the 2030 Agenda for Sustainable Development, providing an opportunity for countries and their societies to embark on a new path aimed at improving the lives of all, while leaving no one behind. This agenda set forth objectives that included eliminating poverty, combating climate change, advancing education, promoting women’s equality, protecting the environment, and redesigning our cities, among other objectives. Goal two, Zero Hunger, established a target for 2030 to eradicate all forms of malnutrition. This included meeting the internationally agreed targets on stunting and wasting in children under five years of age by 2030, as well as addressing the nutritional needs of adolescent girls, pregnant and lactating women, and the elderly [1].

From a conceptual standpoint, malnutrition results from prolonged food deprivation, which occurs when conditions of undernourishment persist over time in a territory. This leads to physiological adaptations in the body necessary for survival; the body goes through acute states that can be fatal or can lead to chronicity and growth arrest, with direct consequences on the proper growth and adequate development of children [2]. Malnutrition can be classified into two types: primary, which is associated with a reduction in food consumption or use (due to scarcity or access difficulties), and secondary, which is linked to an underlying pathology.

In general, primary malnutrition is entirely preventable. In Colombia, the country has the capacity to produce the food necessary for its population [3]. However, not all families have access to the necessary amounts of food to lead healthy lives. Indeed, 28.1% of households experience moderate to severe food insecurity, indicating a compromised quality and variety of food, reduced food quantities, or instances of hunger [4]. Although the prevalence of malnutrition varies depending on the territorial context (as conditions differ between departments experiencing drought or food supply issues compared to those without), wasting is a phenomenon that occurs across all Colombian departments. In 2023, moderate and severe wasting affected children who were born underweight, belonged to low socioeconomic strata, or had mothers with lower educational levels. The national prevalence of moderate or severe wasting in children under five, as reported by health services for the first half of 2023, was 0.31% in Colombia, an increase from 0.27% during the same period in 2022 [5]. This reported prevalence is lower than the actual population prevalence, due to the underreporting of this condition [3,4].

Child malnutrition in Colombia continues to be an alarming issue that profoundly impacts the lives of many children. According to the National Survey of the Nutritional Situation in Colombia (ENSIN) [6], one in ten children experiences stunting, and about two out of every hundred suffer from wasting. This represents around half a million infants nationwide [7]. The root causes of this significant issue extend beyond mere food scarcity and include factors such as poverty, inadequate access to clean water, basic sanitation services, and insufficient parental education [8,9].

The consequences of child malnutrition are multifaceted. Children afflicted with malnutrition not only encounter an increased risk of mortality and susceptibility to diseases such as acute diarrheal disease (ADD) and acute respiratory infections (ARIs), but their growth and cognitive development are also compromised [10,11]. This translates into challenges in learning and full development, which subsequently curtail their future opportunities and perpetuate the cycle of poverty in which they are entrenched [2,12]. Moreover, the Colombian health system incurs significant costs in addressing the complications associated with malnutrition, an aspect that has yet to be analyzed from a societal perspective.

Previous studies have focused on health outcomes or isolated cost factors, whereas our research integrates economic implications with a focus on disability-adjusted life years (DALYs), allowing researchers to better understand the societal and economic impacts of malnutrition on both healthcare systems and households. This approach provides valuable insights for developing comprehensive public health interventions and policy recommendations.

Therefore, determining the burden of disease associated with malnutrition in a country that has the tools and resources to prevent it allows action to be taken and prevents an unjust and avoidable phenomenon from continuing its course. This research seeks to answer the following question: what is the burden of disease attributable to malnutrition in Colombia, and what is its economic impact from a societal perspective?

## 2. Methods

### 2.1. Study Design

A study of the economic burden of this disease was conducted, employing epidemiological methods alongside a cost analysis. For estimating the burden, the methodology utilized in burden-of-disease studies (CES) [13] was adopted to calculate the DALYs (disability-adjusted life years), which were calculated as follows:
DALY=YLL+YLD
YLL: years of life lost to premature death (AD − LE).YLD: years of life lost due to disability (D × PDP = (AD − AO) × DW).
where

LE: life expectancy of the country at the time of the study.

AD: average age of death within the population group.

D: duration of the disability.

AO: age of onset of the disability.

DW: disability weight.

### 2.2. Model

To estimate the DALYs, a Markov model was constructed to simulate the progression of newborn Colombian children, distinguishing between those with and without low birth weight (see Figure 1). The model’s states and the possible transitions between them were developed with input from clinical experts in pediatrics and nutrition who were experienced in nutritional recovery.

This model is commonly used in health decision making to simulate the transitions of a hypothetical cohort between various health states over time. The model assumes that, at the time n, an individual is in the state ${\;X}_{n}$. Each state ${\;X}_{n}$ is mutually exclusive and exhaustive. Consequently, everyone in the model can occupy only one of these states at any given time.

In this case, a discrete-time Markov chain is a sequence of random variables that comply with the Markov property, which implies that the probability of moving to the next state depends only on the current state and not on the previous states. This is as follows:
$$P\left(X_{n+1}=j\;\right|X_{0}=x_{0},\;{\;X}_{1}=x_{1},\;\ldots \;X_{n}=i)=P\left(X_{n+1}=j\;\right|\;X_{n}=i)$$
where
$P\left(X_{n+1}=j\;\right|\;X_{n}=i)$ is the capability of moving from the state i in time n to the state j in time (n+1). The model has an associated transition matrix, $P=\left(p_{ij}\right)$, which includes all the transition probabilities from time n to time (n+1) for moving from state (i) to state j. The model assumes that this matrix is homogeneous in time n and independent of time. Assuming a model with k states, the distribution of the population among them over time can be expressed as the following vector:
$$M^{n}=(M_{1}^{n},\;M_{2}^{n},\;M_{3}^{n},\ldots \;M_{k}^{n})$$
where $M_{k}^{n}$ is the number of individuals that are in the state k in time n. Likewise, the distribution of individuals over time (n+1) is as follows:
$$M^{n+1}=M^{n}*P$$

#### Six-State Markov Model

The model is composed of six (6) states (low birth weight, adequate weight, wasting, stunted height, wasting + stunted height, and death), where death is an absorbent state. The Markov chain was modeled with an initial cohort of 512,611 individuals, equivalent to the Colombian population of live births in 2021 (DANE).

To determine the DALYs, a Markov model was constructed (see Figure 1) in which the probability of becoming ill due to wasting influenced the mortality, chronicity, morbidity, and outcomes of disability associated with it according to its severity. The definition of the variables of the model was determined by consensus with clinical and thematic experts.
Figure 1Markov model.
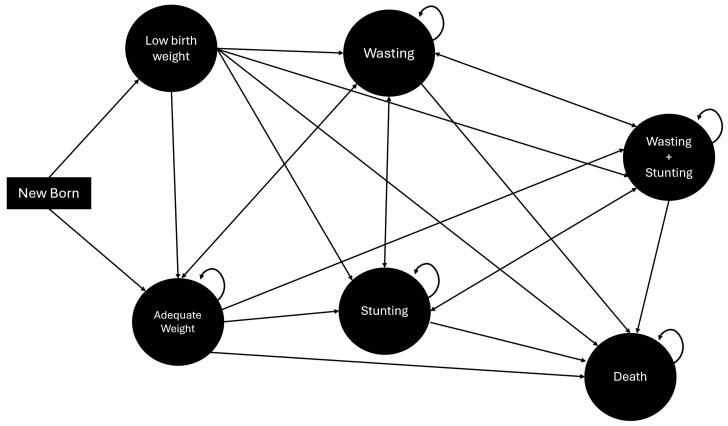


**Adequate Weight****BPN****Wasting****Stunting****DA + RT****Death**RN0.89880.10120.00000.00000.00000.0000Adequate weight0.89320.00000.01200.08200.00100.0118BPN0.76850.00000.02800.15800.02800.0176Wasting0.64000.00000.13820.18900.00700.0258Stunting0.00000.00000.93480.01080.04630.0081DA + RT0.30000.00000.00000.49900.16710.0339Death0.00000.00000.00000.00000.00001.0000

Newborns are classified as either having adequate weight or being underweight. In both conditions, the risk of wasting may occur, which could either resolve or lead to stunted growth. Furthermore, in all scenarios, there is a potential transition to mortality. Like the methodologies employed in burden-of-disease studies, the model adopts life expectancy as the time horizon for calculating the potential years of healthy life lost. However, it is important to note that the estimated burden specifically refers to children under the age of four. The model was estimated using RStudio, version 4.3.1.

### 2.3. Model Parameter Estimation

The demographic and epidemiological parameters were obtained from studies [14,15] and official sources such as the vital statistics of DANE [16], national surveys [17], the RIPS (Individual Records of Health Service Provision) contained in SISPRO [18] (Integrated Social Protection Information System), and the ICBF’s “Cuéntame” mission system. The disability weights for malnutrition were derived from the most recent Global Burden of Disease Study 2019 (GBD 2019), coordinated by the Institute for Health Metrics and Evaluation (IHME) [19]. For transition probabilities not available in the literature, both additive and multiplicative rules of probability were employed, including identification, measurement, and costs valuation. Figure 1 provides a summary of the key parameters used in the model, including the transition probabilities.

#### 2.3.1. Direct Costs

The direct costs associated with the burden of disease were estimated using secondary sources of information [20,21,22]. Priority was given to recent national literature to ensure the relevance and accuracy of the data in the Colombian context. To calculate the costs associated with wasting, the costs associated with acute respiratory infections (ARIs) and acute diarrheal disease (ADD) were used. According to the national literature, the incidence of ARIs and ADD for this age group was 42% and 53%, respectively [23]. Consequently, the model allocated costs in these same proportions. The costs linked to ARIs were computed based on the expenses incurred during an ARI episode. Simultaneously, the costs for ADD were calculated from the unit cost of traditional rehydration therapy combined with the expenses for general hospitalization in a high-complexity medical facility.

The estimation of the costs associated with stunted height, which is assumed to result in delayed cognitive and psychomotor development, assumed that affected individuals, from the age of two up to four years, would require occupational therapy three times per week. Within this framework, it was projected that only one of these three weekly therapy sessions (approximately 33%) would be subsidized by the health system. The remaining two sessions (67%) were expected to be covered by households as out-of-pocket expenses.

For the cost analysis, an annual inflation adjustment was applied to standardize the values to the year 2023. This adjustment ensured that the costs were comparable across different time periods. Table 1 presents the per capita cost for the health system associated with each health state analyzed in the model, along with the respective sources of these data.

#### 2.3.2. Indirect Costs

To estimate the indirect costs, an information collection instrument was developed and administered to caregivers. This instrument gathered data on family income, out-of-pocket expenses, and productivity losses. Initially, a pilot test of the instrument was conducted, leading to the reorganization of some sections and the refinement of term definitions to enhance comprehension. After these modifications, forty-three surveys were conducted from 2023 to February 2024 with the caregivers of children under five years old who were treated for moderate or severe wasting and/or stunting at six hospitals (both public and private) in the Department of Antioquia. To the initially estimated indirect costs, an additional amount was added after two years, representing 67% of the annual costs of occupational therapies, equivalent to USD 920 per year. This additional expense was borne by the households of individuals who exhibited delayed psychomotor development due to stunted growth.

## 3. Results

### 3.1. Burden of Disease

The Markov model estimate indicates that DALYs for malnutrition in Colombia correspond to 419.84 per 1000 inhabitants during the first four years of life. Table 2 and Figure 2 present a summary of the DALY, YLL, and YLD results. The data suggest that the YLL significantly contributes to the increase in DALYs over time.

### 3.2. Direct and Indirect Costs

After reviewing the literature on the direct cost data collection, priority was accorded to studies that provided analyses of the national costs. The subsequent sections detail the costs associated with wasting, particularly those linked to ARIs and ADD. Additionally, the costs related to a height delay are outlined, specifically the expenses for delayed psychomotor development, which include three weekly occupational therapy sessions after the age of 2 years. To ensure comparability among the various cost values, all figures have been adjusted for inflation. This adjustment ensures that the presented costs reflect up-to-date and comparable economic values, facilitating more accurate interpretations.

Table 3 presents the results of the direct and indirect costs per cycle and the cumulative total for the first four years.

The results of the indirect cost analysis revealed that the total reached USD 243,576,094.12 over the four-year period analyzed. Table 3 illustrates the evolution of indirect costs for each cycle.

At the time of the survey, most children were categorized under nutritional recovery, followed by home management with follow-up and re-entry to nutritional recovery. Among the surveyed children, 76.19% were identified as experiencing wasting.

Regarding the demographic details, the children primarily belonged to socioeconomic strata 1, 2, and 3, with a prevalence in strata 1 and 2 (91%). The majority were covered under the subsidized health insurance regime (74.4%), with 23.3% under the contributory regime and 2.3% uninsured. None of the respondents had additional insurance or supplementary plans.

The household characteristics indicated that the largest proportion resided in stratum 1 (55.8%), with 62.8% residing in urban areas. Most households did not include children under 5 years of age, individuals over 65 years of age, pregnant women, or disabled persons (see Appendix A Table A1).

Among the categories of annual out-of-pocket expenses, transportation incurred the highest costs at USD 451.76, followed by food at USD 225.88 and stationery at USD 84.71. When analyzing indirect costs by socioeconomic strata, transportation and food consistently showed the highest expenses across strata 1, 2, and 3. Additionally, the most substantial out-of-pocket expenses occurred during hospitalization, totaling USD 971.89. Detailed figures for these expenditures are available in Table A2, which reveals that the average annual per capita expenditure for households was USD 162.98.

According to the characteristics of the caregivers surveyed, the average age was 37 years. A significant majority (95.35%) were mothers, and 65% were not the heads of their households. The predominant affiliation regime was subsidized (74.4%). The highest level of schooling reported was high school, with 62.8% of caregivers having this educational background. Regarding occupation, 44.19% were engaged in household chores and 25.58% were involved in informal work. Most caregivers did not cease their activities for child care, a situation most prevalent among those engaged in household chores. This finding suggests a lack of support for child care in most cases, as detailed in Table A3.

In terms of the economic characteristics of the households, the average income per household was observed to be USD 352.21, while per capita income averaged USD 190.34. It was noted that most households did not have alternative sources of income.

### 3.3. Economic Burden

The analysis of costs associated with malnutrition in Colombia revealed that the direct costs amounted to USD 128.92 million, whereas the indirect costs tallied up to USD 243.58 million. This resulted in a total cost of USD 372.50 million over the first four years of the cohort’s life. Additionally, the DALYs per 1000 inhabitants were found to be 419.8. Table 4 illustrates the detailed breakdown and evolution of these costs over time.

The primary financial burden was with indirect costs, constituting 65% of the total costs, in contrast to the 35% represented by direct costs. An increasing cost trajectory over time was noted, particularly after the second year of life, when households are required to cover two-thirds of occupational therapy costs as out-of-pocket expenses.

A comparison of these expenditures to the per capita income of individuals right above the extreme monetary poverty line and the monetary poverty line revealed significant disparities. (According to the National Administrative Department of Statistics (DANE), for the year 2022, the per capita extreme monetary poverty line was set at COP 198,698 per month, while the monetary poverty line was established at COP 396,864 per month). Specifically, the average annual out-of-pocket expenditure amounted to 123.4% of the total annual income for individuals right above the extreme poverty line, and 61.8% for those right above the monetary poverty line. Such economic strains suggest that, for households, coping with an income shock resulting from child malnutrition could lead to a higher incidence of monetary and extreme monetary poverty, restricting the household spending flexibility in a manner that perpetuates poverty and deepens existing inequities.

## 4. Discussion

Child malnutrition is a completely preventable public health issue. No child should succumb to this condition, especially in a country with abundant and diverse food resources like Colombia. This problem is linked to health inequities and serves as a catalyst for the loss of human capital and the growth potential of society. Malnutrition contributes to cognitive decline, resulting in a delayed school entry, a poor academic performance, and decreased graduation rates [11,12,24,25]. Poor fetal growth or stunted growth within the first two years of life leads to irreversible damage, including a shorter adult stature, lower educational attainment, reduced adult income, and a lower birth weight in subsequent generations.

This study represents the first comprehensive assessment of the burden of malnutrition from a societal perspective in Colombia. Our findings indicate that the DALYs for child malnutrition in the first four years of life are 419.8 per 1000 inhabitants. Despite high health coverage, a considerable proportion of the associated costs are borne by households: of the total costs attributable to child malnutrition, 32% are indirect costs assumed by households, while 68% are direct costs covered by the health system.

Several studies have shown a correlation between birth weight, subsequent nutritional status, and the development of diseases in adult life [26,27,28]. This evidence highlights the importance of early interventions, particularly before the age of five, as a critical period for improving nutritional outcomes. According to one study, approximately 85% of children born with a very low birth weight and 53% of those born with an extremely low birth weight achieve a normal height by the age of 4. However, children who remain short-statured at age 4 are unlikely to attain a normal adult height [29].

In 2014, the most recent study on the burden of disease in Colombia was published, utilizing data from 2010. It identified a low birth weight as the primary cause of disease burden among children under five years of age, with 134 total DALYs in males and 144 in females. This was followed by asphyxia and birth trauma, accounting for 46 DALYs in males and 48 in females. The study also reported protein-calorie malnutrition, with 4 DALYs in males and 5 in females per 1000 children, and lower respiratory tract infections, with 8.3 DALYs in males and 9.9 in females per 1000 children [14]. Additionally, a descriptive study conducted in Colombia reported the years of life lost due to premature mortality associated with child malnutrition, which ranged from 1162 in 2016 to 6411 in 2019. The years lived with disability varied from 1239 in 2016 to 2257 in 2019, corresponding to 2402 DALYs in 2016 and 8668 DALYs in 2019 [30].

According to the 2016 Global Burden of Disease (GBD) study, diarrheal diseases accounted for approximately 40,125,700 DALYs due to the incidence of diarrhea and associated fatalities among children under five years of age, resulting in about 446,000 deaths (with a range of 390,900 to 504,600) and 1.105 billion episodes (ranging from 962 million to 1.275 billion). After including long-term sequelae associated with malnutrition, diarrhea contributed to an increased total of 55,778,000 DALYs, marking an almost 40% increase (39.0%, with a range of 33.0% to 46.6%) [31].

To mitigate these effects, it is crucial to implement intervention strategies that encompass child nutrition programs and health education, particularly focusing on the first 1000 days of life. This period is critical for optimal brain growth and development [32]. The findings of this study highlight the significance of such interventions: caregivers reported that malnutrition adversely affected both the health of children and the household income, with 38.1% of households experiencing a decrease in income due to child malnutrition.

Although wasting is a notifiable condition in Colombia through its epidemiological surveillance system, SIVIGILA, there are significant gaps in the reporting mechanisms. Notably, stunting does not require mandatory reporting, and the data collected do not fully reflect the population’s reality due to known issues of underreporting. This discrepancy arises because not all medical consultations classify nutritional status, and not all children suffering from malnutrition, whether acute or chronic, seek healthcare services. As a result, population statistics, which are typically reported by five-year surveys, which were conducted in 2005, 2010, and 2015 [17,33,34], should have been utilized. However, since there have been no updates since 2015, data from that year must be used as the most recent. Further research is essential, particularly longitudinal studies that explore the outcomes of child malnutrition, including changes in body composition and other developmental trajectories [35].

It can be inferred from the information reported by the National Institute of Health [5] and the Ombudsman’s Office (Defensoría del Pueblo) for Colombia in 2023 that there is evidence of an increase in the number of cases of child malnutrition, both in its prevalence and in its associated mortality, as there has been a reported increase of 56.3% in infant mortality and 34.9% in cases of moderate and severe wasting [36].

An inverse relationship has been established between public social expenditure and stunting; that is, as public social expenditure increases, the prevalence of stunting decreases. Consequently, it is imperative that governments in Latin America allocate greater budgetary resources to the formulation of comprehensive early childhood care policies. Such policies are essential not only for reducing stunting rates, but also for fostering long-term sustainable development in the region [37].

Within the context of the sociodemographic characteristics of households experiencing wasting, it is crucial to note that the findings of this study reflect the cyclical nature of the wasting scourge, encompassing poverty, hunger, and malnutrition. These conditions are often the result of low-paying jobs or unemployment, low educational levels among caregivers, and inadequate income to ensure food and nutrition security (FNS) in the household [38].

One of the significant challenges in studying the burden of disease for child malnutrition is achieving a level of regional and territorial disaggregation that supports intersectoral decision making at the local level. This is crucial, as addressing the issue requires coordinated efforts across various social determinants of health. Combating child malnutrition in Colombia needs an integrated strategy that incorporates food security policies, nutrition programs, and health education. The data and studies reviewed highlight the need to approach malnutrition not only as a public health issue, but also as an economic and social imperative for the country’s sustainable development. Implementing effective policies and early intervention programs is vital to reducing the burden of malnutrition and enhancing the quality of life for affected children and their households.

Regarding the applicability of this study and practical recommendations to the Colombian government, we propose strategies such as scaling up nutrition programs with a focus on social determination to address root causes, improving malnutrition surveillance systems, and investing in infrastructure. Expanding access to health services in rural areas is also suggested. We will provide technical and financial support to international humanitarian organizations to build capacity for nutrition and health data collection and work with local authorities to implement scalable interventions.

## 5. Conclusions

This study provides a comprehensive analysis of the economic situation regarding the burden of child malnutrition in Colombia from a social perspective, using Markov models to estimate the direct and indirect costs. The results highlight the significant impact of malnutrition on health status and the economic burden it imposes on families and health systems. Based on our findings, we recommend the implementation of targeted nutrition programs and public health policies to mitigate the effects of malnutrition, especially in low-income and rural areas. Future research should focus on extending this model to include long-term effects beyond childhood and studying similar patterns in other Latin American countries to increase generalizability.

## 6. Limitations

Malnutrition is underreported due to gaps in health coverage and data collection, especially in rural areas. Additionally, this is thought to limit potential errors in caregiver-reported data that could affect the accuracy of indirect cost estimates. We also highlight the assumptions used in the cost calculations and their potential impact on the results obtained. Finally, we highlight the limited generalizability of the model outside Colombia and note that, although the model was adapted to the Colombian context, adaptations for other countries are necessary.

## Figures and Tables

**Figure 2 nutrients-16-04302-f002:**
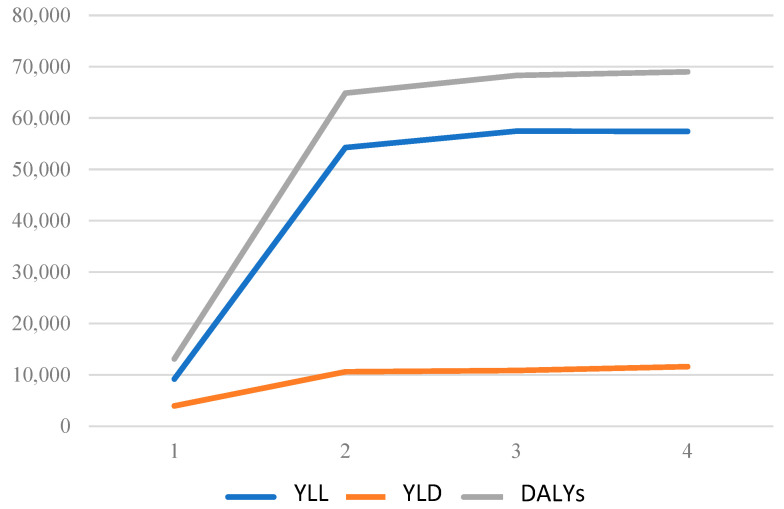
Evolution of years of life lost (YLL), years of life lost due to disability (YLD), and DALYs for the four cycles analyzed by the model.

**Table 1 nutrients-16-04302-t001:** Per capita cost of health states analyzed in the model.

		%	Cost per Capita	Source Year	Cost per Capita 2023	Source	Note
Wasting (W)	ARI	42%	USD 698.16	2019	USD 956.11	Moyano (2019) [20]	The cost of one (1) event.
	ADD	53%	USD 52.60	2015	USD 86.35	Mejía et al. (2016) [21]	The unit cost of rehydration treatment, combined with general hospitalization for a high-complexity service, was calculated.
Stunting (S)	Delay in Cognitive and Psychomotor Development	100%	USD 560.52	2021	USD 460.03	Ministry of Health (2021) [22]	Equivalent to 33% of occupational therapy costs, three times a week.
(S + W)	(S + W)	100%	.	.	USD 956.11	Assumption	It was assumed to be equal to the maximum cost between the previous items.

**Table 2 nutrients-16-04302-t002:** Years of life lost, years of life lost due to disability, and DALYs for each cycle.

t	YLL	YLD	DALYs	DALYs per 1000 Habitants
1	9161.57	3942.97	13,104.54	25.56
2	54,245.38	10,587.72	64,833.10	126.48
3	57,471.01	10,820.21	68,291.22	133.22
4	57,394.77	11,593.28	68,988.04	134.58
Total	178,272.73	36,944.17	215,216.89	419.84

**Table 3 nutrients-16-04302-t003:** Direct and indirect costs of child malnutrition for a cohort of 512,611 children over four years (USD, 2023).

Time (t)	Direct Costs	Indirect Costs
T = 1	USD 5,696,701.18	USD 9,064,985.88
T = 2	USD 26,443,277.65	USD 15,305,211.76
T = 3	USD 47,409,712.94	USD 107,372,367.06
T = 4	USD 49,369,400.00	USD 111,833,529.41
Total	USD 128,919,089.41	USD 243,576,094.12

**Table 4 nutrients-16-04302-t004:** Total direct and indirect costs per model cycle (USD, 2023).

Time (t)	Direct Costs	Indirect Costs	TOTAL
T = 1	USD 24,210.98	USD 38,526.19	USD 62,737.16
T = 2	USD 112,383.93	USD 65,047.15	USD 177,431.08
T = 3	USD 201,491.28	USD 456,332.56	USD 657,823.84
T = 4	USD 209,819.95	USD 475,292.50	USD 685,112.45
Total	USD 547,906.13	USD 1,035,198.41	USD 1,583,104.54

## Data Availability

The consolidated survey data are in Appendix A Table A1, Table A2 and Table A3.

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
