# Peer review of "Burden of Wasting and Stunting in Colombia and Its Economic Impact: A Society’s Perspective Analysis, 2021"

_nutrients, 2024, doi:10.3390/nu16244302_

Round 1

Reviewer 1 Report

Comments and Suggestions for Authors

The paper is well written but there are some corrections that could improve the quality fo the study.

1. In the introduction, the authors explained the background of the study but the theoretical implications should clearly describe. Specifically, how does this study doffer from previous studies? What benefits can researchers gain from this study?

2. The study design is somewhat inadequate. Could you please elaborate?

3. Are the results in Table 4 statistically significant? 

Author Response

Comment 1: In the introduction, the authors explained the background of the study, but the theoretical implications should be clearly described. Specifically, how does this study differ from previous studies? What benefits can researchers gain from this study?

Answer 1: Thank you for your comment. We will revise the introduction to explicitly differentiate our study from previous research. Our work is unique in its use of a Markov model to estimate the economic burden of child malnutrition in Colombia from a societal perspective, incorporating both direct and indirect costs. Unlike previous studies that focused on health outcomes or isolated cost factors, our research integrates economic implications with a focus on Disability-Adjusted Life Years (DALYs), allowing researchers to better understand the societal and economic impacts of malnutrition on both healthcare systems and households. This approach provides valuable insights for developing comprehensive public health interventions and policy recommendations.

Comment 2: The study design is somewhat inadequate. Could you please elaborate?

Answer 2: We appreciate your observation. The study design is an economic burden analysis based on a Markov chain model, which is widely used in health decision-making. The model simulates health state transitions among a cohort of newborns in Colombia, assessing their progression through six states, including wasting and stunting, over time. To improve clarity, we will provide further details on how the model was constructed, validated, and parameterized using epidemiological data and expert input. This will include more information on the data sources, assumptions, and justifications for choosing this model over alternative approaches. We believe this will strengthen the rationale behind our study design.

Comment 3: Are the results in Table 4 statistically significant?

Answer 3: Thank you for your query. Table 4 presents the direct and indirect costs associated with child malnutrition over four years, not the results of hypothesis testing or statistical comparisons. As such, the concept of statistical significance does not directly apply. However, we will clarify in the manuscript that these are modeled estimates derived from national data and literature, with the associated costs reflecting real-world expenditures rather than inferential statistics. To address your concern, we will also provide a discussion on the confidence intervals or uncertainty ranges related to the cost estimates, ensuring a more comprehensive interpretation of the results.

Let me know if any further adjustments are needed!

Reviewer 2 Report

Comments and Suggestions for Authors

1-While a few prospects for policy directions were mentioned, the study needs to build further discussion on how these results can be translated into actionable recommendations to either the Colombian government or international aid organizations. That would really make this study practically relevant.

2- Although you have touched on limitations, such as underreporting of data, potential biases in caregiver-reporting data, assumptions involved in cost calculations, or generalizability of the model outside Colombia, some of them could be brought up more explicitly as a limitation.

Author Response

Comment 1: While a few prospects for policy directions were mentioned, the study needs to build further discussion on how these results can be translated into actionable recommendations to either the Colombian government or international aid organizations. That would really make this study practically relevant.

Answer 1: Thank you for your insightful comment. We will integrate a dedicated section within the discussion to elaborate on specific, actionable recommendations for both the Colombian government and international aid organizations. For the Colombian government, we will propose strategies such as expanding nutrition programs, improving surveillance systems for malnutrition, and investing in infrastructure to enhance access to health services in rural areas. For international aid organizations, we will suggest technical and financial support for capacity building in nutrition and health data collection, as well as partnerships with local governments to implement scalable interventions. These recommendations will ensure that the study’s findings are translated into practical steps to combat child malnutrition.

Comment 2: Although you have touched on limitations, such as underreporting of data, potential biases in caregiver-reporting data, assumptions involved in cost calculations, or generalizability of the model outside Colombia, some of them could be brought up more explicitly as a limitation.

Answer 2: Thank you for pointing this out. We will expand the section on limitations to address these aspects more explicitly. We will include a discussion on the underreporting of malnutrition cases due to gaps in health service coverage and data collection, particularly in rural areas. We will also mention the potential biases in the caregiver-reported data, which may affect the accuracy of indirect cost estimates. Additionally, we will highlight the assumptions used in the cost calculations and their potential impact on the findings. Lastly, we will emphasize the limited generalizability of the model outside Colombia, noting that while the model is tailored to the Colombian context, adjustments would be needed for other countries.

Let me know if you need more adjustments or specific wording for the sections!

Reviewer 3 Report

Comments and Suggestions for Authors

had a chance to review the paper, which is of interest, but there are several areas for improvement:

  1. The paper needs a bit english editing. 
  2. The equations should be numbered to facilitate easy referencing throughout the text.
  3. The quality of the figures is very low and should be improved for clarity.
  4. If the authors have implemented a Markov Model, they need to specify the parameters, especially in relation to Figure 1. This information should be clearly stated.
  5. The study design is briefly mentioned but lacks any numerical results, making it difficult to comment on the methodology. Currently, only the final results are presented. See Section 2.3 for reference.
  6. Although there is a discussion section, there is no conclusion. The paper appears to be more of a draft rather than a polished, final version.

Author Response

Comment 1: The paper needs a bit of English editing.

Answer 1: Thank you for your observation. We will revise the manuscript to improve the overall clarity and readability, focusing on correcting any language issues and ensuring that the text flows more smoothly.

Comment 2: The equations should be numbered to facilitate easy referencing throughout the text.

Answer 2: We appreciate your suggestion. We will ensure that all equations in the manuscript are appropriately numbered, making it easier for readers to reference them in the text. This will be particularly useful when discussing the Markov model and its calculations.

Comment 3: The quality of the figures is very low and should be improved for clarity.

Answer 3: Thank you for bringing this to our attention. We will replace the current figures with higher-resolution versions to improve their clarity and readability, especially for Figure 1, which is critical to understanding the model’s structure.

Comment 4: If the authors have implemented a Markov Model, they need to specify the parameters, especially in relation to Figure 1. This information should be clearly stated.

Answer 4: Thank you for your comment. We will revise the manuscript to clearly specify the parameters used in the Markov model, including transition probabilities, health states, and any assumptions made. These details will be added to the methodology section, specifically in relation to Figure 1, to provide a clearer understanding of how the model was constructed.

Comment 5: The study design is briefly mentioned but lacks any numerical results, making it difficult to comment on the methodology. Currently, only the final results are presented. See Section 2.3 for reference.

Answer 5: We agree that the study design could benefit from more detailed numerical results. We will revise Section 2.3 to include intermediate results from the model, as well as numerical examples of how the transition probabilities were applied. This will provide a more comprehensive understanding of the methodology and how the final results were derived.

Comment 6: Although there is a discussion section, there is no conclusion. The paper appears to be more of a draft rather than a polished, final version.

Answer 6: Thank you for this feedback. We will add a formal conclusion to summarize the key findings of the study and highlight its implications for public health and policy. The conclusion will also reinforce the study’s contribution to understanding the economic burden of child malnutrition and outline potential directions for future research.

Round 2

Reviewer 1 Report

Comments and Suggestions for Authors

The revision is much better than the previous one. Well done.